

# Characterizing anomalous geomagnetic induction from coastal effects with transfer functions and gradient measurements

András Attila Csontos[1]

[1] HUN-REN Institute of Earth Physics and Space Science, Sopron, Csatkai E. utca 6-8., Hungary

5 *Correspondence to*: András Attila Csontos (csontos.andras@epss.hun-ren.hu)

**Abstract.** The occurrence of anomalous subsurface currents in a region is of significant geophysical importance. Several geomagnetic methods have been developed to characterize the effects of geomagnetic induction. Typically, the intensity and direction of the inducing processes are determined using simplified transfer functions that relate corresponding horizontal and vertical geomagnetic components. The geomagnetic field associated with nearby anomalous currents is expected to be inhomogeneous. Additionally, the distortion of the measured geomagnetic field's geometrical structure, as indicated by the magnetic gradient, is a relevant parameter in this context. In this comprehensive study, both methods are applied to simultaneous measurements of geomagnetic induction and magnetic gradient conducted at two geomagnetic repeat stations near the Adriatic Sea. Furthermore, a novel concept for a magnetic gradiometer is introduced. A strong correlation was observed between the direction and intensity of the calculated Parkinson vectors and the strike directions and intensities of the horizontal geomagnetic gradient during periods influenced by the seaside effect. The primary conclusion is that geomagnetic gradient measurements are highly effective for characterizing the effects of geomagnetic induction and/or quasi-stationary subsurface currents.

## 1 Introduction

The geomagnetic field is always a superposition of magnetic influences of different sources. One of the main task of the geomagnetic observation is to identify and separate the contribution of the various sources present in the geomagnetic record.

From the perspective of origin, geomagnetic fields can be broadly classified into internal and external components. The source of the external field lies outside the solid Earth. The internal field consists of two components: the main field, generated in the Earth's core, and the crustal field, which includes lithospheric contributions and magnetic fields resulting from electric currents caused by various processes (e.g., telluric currents of geochemical or hydrological origin, or internal geomagnetic induction effects) (Cull et al. 1986; Viljanen et al. 1995). This study aims to investigate the correlation between geomagnetic gradients and induced current systems observed near coastal repeat stations, using gradient measurements and transfer function analysis.



The geomagnetic absolute measurement, used in the article, are routinely used in the geomagnetic observatory practice, and therefore can be an easily accessible aid for specialists in detecting geomagnetic induction appearing in measurements. Identifying the effect of geomagnetic induction helps the observer in interpreting the measurement results and in more accurately fulfilling the observatory's purpose (Jankowski and Sucksdorff 1996).

Geomagnetic induction:

At mid-geomagnetic latitudes, the external fields originate from distant (more than 100 km away) current systems in the ionosphere and magnetosphere. Due to the large distance of these external currents, the resulting geomagnetic field is relatively uniform on the Earth's surface at mid-latitudes. However, near the equatorial and polar regions, the equatorial and auroral electrojet produce inhomogeneous external fields, necessitating additional considerations (Arora et al. 1999; Cull 1985). During strong geomagnetic storm the auroral electrojet can appear in mid-geomagnetic latitudes also.

Variations in the external field appear on the Earth's surface as electromagnetic (EM) waves, ideally in the form of plane waves. Depending on the region, we can distinguish between normal and anomalous points, based on the heterogeneity of the Earth's electrical conductivity structure. To simplify the analysis, the geomagnetic record can be divided into the sum of a constant (DC) field and a fluctuating (AC) field. In this framework, only the AC field, driven by external field variations, contributes to electromagnetic induction within the Earth.

At a normal point (i.e., a homogeneous half-space or a horizontally layered, laterally homogeneous geological formation), EM waves penetrate the crust to a depth known as the skin depth, given by:

$$\delta_S = \left( \frac{2}{\omega \mu_0 \mu_r \sigma} \right)^{\frac{1}{2}}, \tag{1}$$

where $\omega$ is the angular frequency of the wave and $\sigma$ is the conductivity, $\mu_r$ is the magnetic permeability of the medium. We can conclude that the EM waves reach different penetration depth on different frequency. Since $\mu_r \approx 1$ for most sediments, mainly the local conductivity distribution modulates the penetration of a certain EM frequency . The EM wave induces eddy currents in the crust, which dissipate as thermal energy in conductive sediments.

1) The presence of eddy currents at a normal point results in only a very weak vertical magnetic field variation, as no persistent correlation exists between the vertical and horizontal geomagnetic components. When the conductivity is low, the the EM wave remains approximately identical in the sediments. However, in the case of high conductivity, the horizontal geomagnetic components increase, and phase shifts between external EM wave and induced horizontal magnetic fields become observable due to the eddy currents.

2) The eddy currents in the Earth's stratified interior produce only a mirror image of the external EM wave if the conductivity of the upper layer is very high, implying that the geometrical structure of geomagnetic field lines remains stable.

Such normal points are ideal locations for geomagnetic observatories because they allow clearer separation of internal and external geomagnetic variations.





In light of the above, we define an anomalous point as a location within a region that is laterally intersected by an electrically distinct area. The conductance contrast between the adjoining zones remains high. Near the interface of these regions, magnetically induced eddy currents flow along a band at the conductivity boundary where the conductance is higher. As a result, a relatively stable, horizontally oriented anomalous current system forms parallel to the conductivity contrast.

In this situation, for several frequencies, the horizontal variation of the geomagnetic field—particularly its component

perpendicular to the conductivity boundary—contributes to sustaining the anomalous current system. These currents induce a significant vertical geomagnetic field at the surface, whose variations show opposing changes on either side of the boundary. The fluctuation of the vertical geomagnetic component (Z) mimics those of the horizontal components (X and Y). The presence of an intensive Z variation is one of the indicators of the induction effect. Due to the coupling between anomalous Z variation and horizontal field variation, the direction of the anomalous current system can be determined. See

the details in Figure 1.

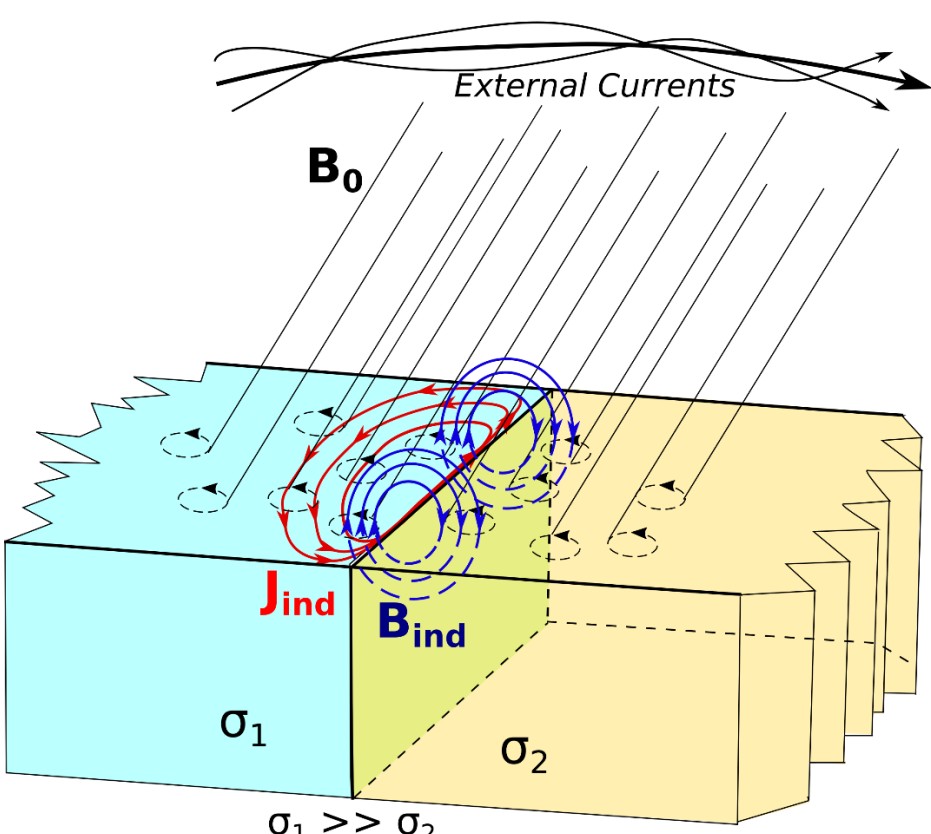

**Figure 1.: Schematic illustration of electromagnetic induction at a lateral conductivity boundary due to an external plane wave. The left region (blue) represents a highly conductive material, while the right region (yellow) has effectively zero electrical**

**conductivity. The Earth's main magnetic field (B₀) varies laterally under the influence of external current systems. The resulting**





**induced currents ($J_{ind}$) flow primarily near the conductivity boundary (green surface) and are shown by red loops. These currents generate a secondary induced magnetic field ($B_{ind}$), illustrated by the blue loops. The model highlights the enhanced induction effect and current concentration caused by conductivity contrast.**

The correlation between vertical and horizontal components can be quantified due to the linear nature of Maxwell's equations. The reliability of such calculations improves if a normal point is used as a reference site (Viljanen et al. 1995; Arora et al. 1999). Details of the robust, simplified complex transfer function determination are presented in Section 3.1. Conversely, the geomagnetic field structure induced by anomalous currents in the Earth is both inhomogeneous and time-varying. Short-term fluctuations in the differences between corresponding geomagnetic elements are a unique signature of

such anomalous currents (Csontos et al. 2012). This characteristic distinguishes the induced magnetic field (due to subsurface currents) from all other sources of geomagnetic variation. In such regions, the effect of anomalous currents can be detected via localized fluctuations in the differences between corresponding components of the geomagnetic field (Csontos 2013). This phenomenon results from the transformation of the geometrical structure of the geomagnetic field. Therefore, a proper analysis requires examining geomagnetic gradients. More theoretical details on geomagnetic gradients are discussed

in (Pedersen and Rasmussen 1990; Eötvös 1998).

The measurement of geomagnetic gradients places focus on the magnetic effects of anomalous currents at the boundaries of two geologically distinct conducting units. (In such cases, the contributions of both the external field and the uniform internal field become negligible.)

The appearance of anomalous currents channels a significant portion of the external EM wave energy into inhomogeneous,

induced geomagnetic fields. The location of the induced field's source—the anomalous current—is thus confined to a relatively narrow band of the shore. These conditions lead to two important consequences:

1. Near the boundary between sediments with different physical (magnetic and electrical) properties, the energy density of the geomagnetic field becomes relatively high. This implies the presence of geomagnetic gradients evident.

2. According to the general principle of energy conservation, enforced by Lenz's law, the direction of the induced magnetic

field must generally oppose the changes in the external horizontal geomagnetic field. As a result, the most pronounced inhomogeneous induced fields are expected to occur in the horizontal plane, directed opposite to the change of the external (primary) magnetic field. Fluctuations in the induced EM field propagate as secondary EM waves during the effect.

It should be noted that the phenomenon of geomagnetic induction can occur not only in the case of lateral conductivity contrast, but can also be created by, for example, ocean currents (Irrgang et al. 2016), subsurface fluid flows (Woods and

Lilley 1980), etc. The aim of this study is to examine the 2D case that has been studied several times and outlined above, so we will limit ourselves to this.

In a previous article (Csontos 2013), a new observatory practice was proposed for identifying the effects of induced current systems based on fluctuations in local differences between corresponding components of the geomagnetic field.



Fluctuations in the baseline values of a calibrated variometer were interpreted as fluctuations in the local differences between
geomagnetic components. This interpretation was supported by two key observations:

- Unstable spatial geomagnetic differences were confirmed by fluctuating values in the differences between simultaneous total field records, which are absolute measurements (Csontos 2013).

- No similar fluctuations were detected in measurements conducted at an inland station, and the characteristics of the fluctuations could not be attributed to any known environmental influences.

It was demonstrated that records from a Declination-Inclination Magnetometer (DIM)—which measures the gradient of the horizontal magnetic field—can be used to correct fluctuations observed in the differences between two total field instruments. These measurements were performed by independent devices, and the total field records were based on absolute
measurements. The successful correction confirmed that the fluctuations in the local differences between total field records resulted from variations in the horizontal geomagnetic field gradient. Accordingly, a new observatory practice was introduced to identify the influence of time-varying geomagnetic gradients and distinguish them from environmental effects (e.g., temperature variations) (Csontos 2013).

Subsequently, a case study was published (Vujić and Brkić 2016) based on the same repeat station measurements,
confirming the presence of a sea-side effect during two re-occupations of a repeat station in Croatia.

In this article, a comprehensive analysis is presented using data from the two aforementioned repeat stations located near the Adriatic coast. The results of Parkinson vector determinations (Vujić and Brkić 2016) are compared with geomagnetic gradient vectors measured in the horizontal plane. To enable this comparison, a method is introduced that utilizes DIM instrumentation to measure horizontal gradients of geomagnetic variation.

The measurement of geomagnetic gradients can serve not only for mapping the dipole anomaly of the crust (Pedersen and Rasmussen 1990) but also for identifying and characterizing the effects and development of subsurface current systems.

Possible sources of anomalous currents within the crust or lithosphere include hydrogeological processes (Woods and Lilley 1980), geochemical processes (Cull 1985), or piezoelectric effects. Additionally, the presence of high-conductivity fluids in subduction zone lithosphere has been repeatedly confirmed (Han et al. 2021).

**2 How to measure the direction of the horizontal gradient of the geomagnetic field using a DIM device?**

In the introduction, we note that a deeper understanding of the chapter requires knowledge of the process of absolute geomagnetic measurements. To learn about this, among many other very good options, the previously published open access article (Csontos and Šugar 2024) may be one of the best choices, the topic of which is precisely the analyzed measurement series with the same notation system. Furthermore, all measurement protocols are also available in digital form (Csontos
2023). With such a level of transparency of measurement data, measurement procedures and previous studies, it seems that



detailed disclosure of previously published knowledge can be omitted. In this study, we will focus on the connections that are closely related to the current topic.

The Declination-Inclination Magnetometer (DIM) instrument is widely used for performing absolute declination and inclination observations. The procedure is based on the specific features of a non-magnetic theodolite equipped with a one-component fluxgate magnetometer. The operation of the DIM instrument was first described by Lauridsen in 1985, and since then, numerous articles have been published presenting new findings from tests and developments aimed at improving modern geomagnetic observatory practices (Marsal and Torta 2007; Brunke and Matzka 2018; Csontos and Šugar 2024).

The instrument was originally designed for use in a uniform geomagnetic field, although a few tests have also been conducted under inhomogeneous conditions (Gilbert and Rasson 1998). These experiments primarily aimed to assess the device's tolerance to magnetic impurities. One of the conclusions was that while the offset of the D&I magnetometer is highly sensitive to magnetic contamination, this sensitivity does not affect the final result of the observation, provided that only the telescope of the instrument contains magnetic materials.

An alternative approach has also been introduced, based on the observation that the readings of the D&I magnetometer are not obtained from a single point, but rather from a small spatial volume (Csontos et al. 2012; Csontos 2013; Csontos 2019). Typically, readings are taken at eight discrete positions during a complete set of absolute measurements. The sampling locations span a maximum diameter of eight centimeters, which justifies the assumption that the geomagnetic field changes linearly within this small volume — an assumption that aligns with the "magnetic hygiene" standards of geomagnetic observatories. This linearity enables the application of the geomagnetic gradient approach.

**2.1 Magnetic gradients measurements calculated from inclination samples of DIM instrument**

Inclination readings can be defined as follows:

$$V1 = Nup, V2 = Sdn, V3 = Ndn, V4 = \text{Sup}, \qquad (2)$$

where, for example, Nup denotes the telescope pointing north (geomagnetic north) with the sensor in the upper position.

The inclination can be determined using the following equations:

$$I = (V1 + V2 - V3 - V4)/4 + 90° \qquad (3)$$

$$I_1 = (V1 - V4)/2 + 90° \qquad (4)$$

$$I_2 = (V2 - V3)/2 + 90° \qquad (5)$$

Where (4) and (5) are sufficient but (4) is measured on the telescope and (5) is behind the telescope.

The difference between these two inclination determinations can be interpreted as the inclination difference between the upper and lower observation points. This difference is attributed to the curvature of the geomagnetic field within the magnetic meridian plane. Since these values are derived from absolute measurements, the resulting difference is also absolute. Consequently, this curvature indicates the presence of a gradient vector in the meridional plane that is perpendicular to the total magnetic field vector (Csontos 2019).



The intensity of this gradient vector ($\mathbf{B_m}$) along the z-axis, which is aligned with the direction of the total field, can be calculated as:

$$\gamma = \frac{1}{F}\frac{\delta \boldsymbol{B_m}}{\delta z} \tag{6}$$

where $\gamma$ is the measured inclination difference and F is the magnitude of the total magnetic field. This expression yields the rate of directional change of the total field in the meridional plane. If z is oriented vertically, then this value represents a vertical derivative.

To meet the objectives of the present study, a deeper understanding of horizontal gradient measurement using the DIM device is also necessary.

The offset of the D&I fluxgate magnetometer can be determined in multiple ways based on inclination readings, such as:

$$S_1 = \big(V1 - (V2 - 180)\big) \cdot F \cdot \pi/360, \tag{7}$$

$$S_2 = \big((180 - V4) - (360 - V3)\big) \cdot F \cdot \pi/360. \tag{8}$$

In principle, the offset should be a scalar value and therefore independent of the positions at which the measurements are taken. However, earlier studies have demonstrated that the measured offset is highly sensitive to inhomogeneities in the magnetic field or to magnetic impurities in the DIM device [12]. These studies focused primarily on remanent magnetization within the D&I sensor and the theodolite, which were considered to be sources of internal magnetic gradients.

In contrast, the present approach emphasizes the geometrical structure of the ambient geomagnetic field—an aspect that remains independent of the instrument's construction or magnetic contamination.

Assuming equations (4) and (5) yield equal values, any discrepancy between the offsets computed from equations (7) and (8) suggests the presence of non-parallel magnetic field lines within the measurement volume. The DIM instrument can detect this condition if the "magnetic center" of the probe is displaced from the theodolite's vertical axis. In such a case, the magnetometer's zero outputs (e.g., V1 and V2), which correspond to magnetic field measurements taken with opposite sensor polarities, will be spatially asymmetric with respect to the axis of rotation. This is illustrated in Figure 2.




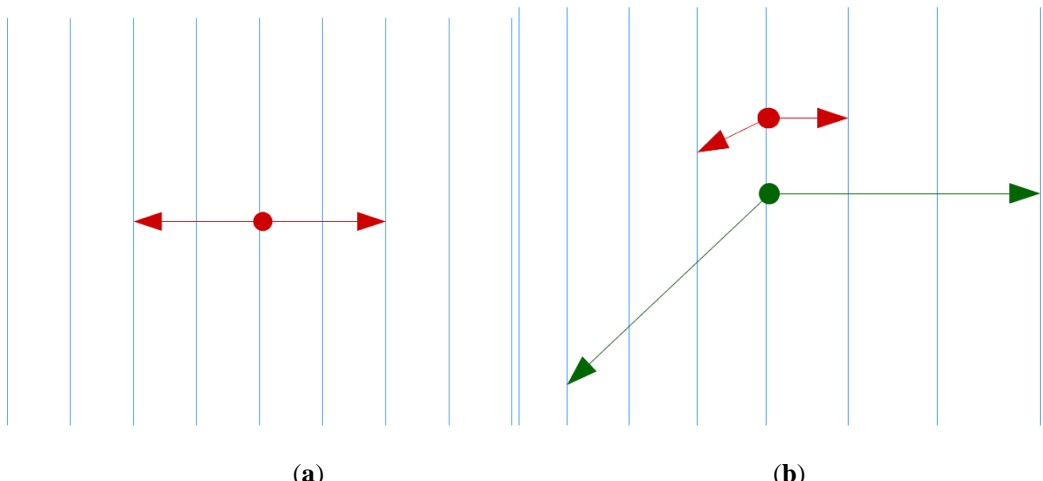

190          **(a)**          **(b)**

**Figure 2: (a) The figure depicts measurements in a homogeneous magnetic field, with blue lines representing equipotential surfaces. Arrow endpoints show zero-output positions, and arrow directions represent sensor polarities. In this configuration, the**
**offset calculated using Equation (7) is zero. (b) Measurements are shown in an inhomogeneous magnetic field using a sensor with zero intrinsic offset. However, the calculated offset based on Equation (7) is non-zero because the angle difference between opposing measurements is not 180°. Here, the dependence of the offset on the distance between the rotation axis and the actual measurement point is clearly visible (see green vs. red arrows).**

The discrepancy between measured and true offsets is proportional to the magnetic gradient in the meridional plane. The
observed gradient direction is perpendicular to the total magnetic field vector, which represents a directional derivative. If

the total field vector is vertically oriented, this indicates a horizontal derivative component.

Although the curvature of the geomagnetic field and the directional derivative of the gradient are conceptually independent,

both quantities can be derived from inclination measurements.

**2.2 Magnetic gradients measurements calculated from declination samples of DIM device**

The declination readings are:

$$A_1 = Eup, A_2 = Wup, A_3 = Edn, A_4 = Wdn, \qquad (8)$$

where, for example, Eup refers to the configuration in which the telescope is oriented toward the East (geomagnetic East),

the optical axis of the telescope is horizontal, and the DI sensor is positioned above the axis.

The computation of declination involves the following quantities:

$$D = (A_1 + A_2 + A_3 + A_4)/4 - (B - A_Z), \qquad (9)$$

$$\delta = (A_3 + A_4 - A_1 - A_2)/4, \qquad (10)$$

$$\varepsilon_D = (A_1 - A_2 - A_3 + A_4 \pm 2\pi)/4 \cdot tanI, \qquad (11)$$

$$S_{0D} = (A_1 - A_2 + A_3 - A_4) \cdot H/4 \cdot 180/\pi. \qquad (12)$$

Where $A_Z$ is the direction of azimuth mark, B is the result of mira readings, (B-$A_Z$) is the azimuth correction factor, D is the
computed magnetic declination, δ denotes the horizontal-plane misalignment of the sensor, $\varepsilon_D$ is the sensor's vertical





misalignment, derived from declination data, $S_{0D}$ is the offset value obtained from declination readings, H is the horizontal intensity of the geomagnetic field.

According to the definition of $S_{0D}$ parameter, the values corresponding to opposite orientations of the fluxgate probe are used. Consequently, the geometrical interpretation presented in Figure 2 (in Section 2.1) applies here as well; see also (Csontos 2013) for further methodological details.

It should be noted that the presence of the magnetic gradient is reflected not only in the offset value but also in the magnitude of the calculated misalignment errors. The two quantities show similar changes if we examine them in the same dimension, e.g. angle or nT. The misalignment errors of the sensor can be influenced by mechanical changes, therefore in this study we only examine the changes in the offset in order to rely on the most reliable quantity. In stabilized environment of geomagnetic observatory, the change of both parameters can be a consequence of the gradient instability. Further experiments need to characterise this approach.

The discrepancy between the measured and actual offset values derived from declination readings is proportional to the horizontal magnetic gradient vector in the observation volume. This gradient vector lies perpendicular to the direction of geomagnetic North.

## 2.3 Accounting for the sample distances

As demonstrated in Figure 2, the distance between the sampling points corresponding to null readings is identified as a key parameter in interpreting offset results. In the case of declination measurements, the situation is relatively straightforward. Since the optical axis of the theodolite remains horizontal, the calculated offset is directly proportional to the horizontal component of the geomagnetic gradient in the geomagnetic East–West direction. Accordingly, the effective distance can be identified with the distance between the "magnetic centers" of the sensor in positions A1 and A2 positions, denoted as $d_D$.

The hypothesis that horizontal magnetic gradients also contribute to the offsets observed in inclination measurements requires more nuanced discussion. Based on the studied site, it is reasonable to assume that the dominant direction of the geomagnetic gradient lies in the horizontal plane. However, during inclination observations, the telescope's optical axis is tilted, and thus the readings V1 and V2 are not taken along a strictly vertical path. Consequently, the effective distance between these two measurements depends on the measured geomagnetic inclination. Given the geometrical arrangement described above, the effective distance ($d_{eff}$) between the V1 and V2 readings can be expressed as:

$$d_{eff} = d \cdot cos(I), \tag{13}$$

where d is the physical distance between the sensor position V1 and V2 and I is the local geomagnetic inclination. See Figure 3. for the schematic representation.





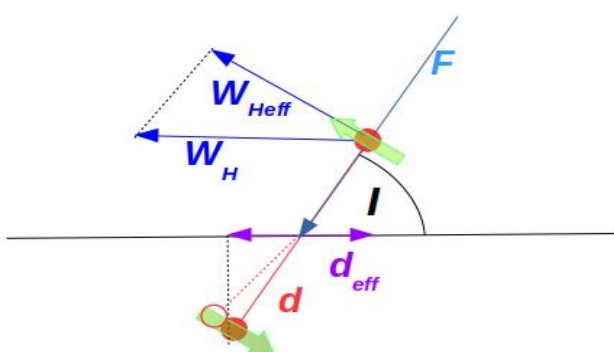

**Figure 3.: Measurement of a horizontal derivative using a tilted probe. The figure presents a side-view perspective, perpendicular to the geomagnetic meridian plane. The green arrows indicate the positions and polarities of the DI fluxgate sensor during inclination readings. The red dots represent the sensor positions at V1 and V2 separated by distance d. The total geomagnetic field vector is denoted by F, with inclination I. The blue horizontal vector ($W_H$) represents the hypothetical geomagnetic gradient. Its effective component along the sensor axis, $W_{Heff}$, is also shown. The effective distance between V1 and V2 samples is $d_{eff}$ (purple).**

**The red open circle shows the expected null-point of the DI output if influenced by the horizontal gradient $W_H$.**

The direct measurement of the distances ($d_D$) and d is limited in accuracy. In the scope of the present study, exact calibration of the gradient measurements is not attempted. Instead, reduced offset values are calculated, which are proportional to the actual geomagnetic gradients. For this purpose, only the ratio of the effective distances is required.

This simplification is supported by two assumptions:

  1. The linearity of trigonometric functions in the case of small deflection angles (i.e., $\sin(x) \approx x$ for small x),

  2. The presumption that the geomagnetic gradient is approximately constant within the small measurement volume formed by the eight discrete DIM sample points.

**2.4 Determination of the real offset**

The evaluation described above can only be considered complete if the difference between the real and the measured offset values is taken into account. From a theoretical perspective, the real offset of the DIM instrument is assumed to be a scalar quantity, typically within the range of a few nanoteslas. The origin of this offset—whether instrumental, structural, or environmental—is immaterial, but its temporal stability is considered a prerequisite for reliable absolute geomagnetic measurements, as previously suggested in Lauridsen 1985.

At the current stage of instrumental development, it can be assumed that the short-term stability of the offset (on the scale of a few days) is sufficiently ensured. Nonetheless, slight variations due to temperature sensitivity cannot be entirely ruled out.

In the presence of inhomogeneous magnetic fields, the measured offset may be affected by local field gradients, thereby obscuring the true, intrinsic value of the instrument's offset. This makes direct determination of the real offset under such conditions inherently uncertain.

However, several methods remain available to gain insight into the real offset:





• The most straightforward and reliable method is to perform an absolute measurement in a uniform geomagnetic field. In such a scenario, the offset can be calculated as part of standard observatory procedures. For example:

$$S_0 = -[(V_1 - V_2 - V_3 + V_4)/4 + 90] \cdot F \cdot 180/\pi. \tag{14}$$

• Another valid approach is to carry out the measurement inside a zero-field environment, such as in a shielded laboratory space.

Further alternative techniques for real offset determination are discussed in Section 4 of this paper.

**2.5 Summary of magnetic gradient measurement procedures**

Based on previously published findings [6], it has been established that the difference between the measured offset derived from declination observations and the real offset value is proportional to the horizontal geomagnetic gradient in the geomagnetic East–West direction, i.e. perpendicular to the geomagnetic meridian plane. This component of the gradient is

denoted as $S_{Dc}$.

$$S_{Dc} = S_D - S_0$$

Where $S_D$ is the measured offset obtained from the declination observation, and $S_0$ is the real offset.

In the case of inclination measurements, the measured offset ($S_I$) is also influenced by magnetic gradients, and similar corrections are required. As discussed in Section 2.3, two additional correction factors must be considered: one for distance, and one for directional projection. The corrected value $S_{Ic}$ is defined as:

$$S_{Ic} = \frac{S_I - S_0}{p \cdot sin(I)}, \tag{16}$$

where $p$ is the ratio of the effective distances between sensor positions with opposite polarity in inclination and declination configurations. The magnitude of the horizontal geomagnetic gradient ($W_H$) can then be expressed as proportional to the vector sum of $S_{Dc}$ and $S_{Ic}$.

$$W_H = \sqrt{S_{Dc}^2 + S_{Ic}^2}$$


$$. \tag{17}$$

Furthermore, the strike direction of the horizontal geomagnetic gradient ($Az_W$) can be determined using the measured declination $D$, taking into account that ($S_{Ic}$) lies within the meridional plane, while ($S_{Dc}$) is perpendicular to it:

$$Az_W = \arctan\left(\frac{S_{Dc}}{S_{Ic}}\right) - D, \tag{18}$$

,

This formulation provides both the magnitude and orientation of the observed horizontal magnetic gradient vector, as

derived from the analysis of inclination and declination readings performed with the DIM instrument.





## 3 Repeat station measurements near the Adriatic Sea and the main findings

During the summer of 2010, repeat station (RS) measurements were conducted near the Adriatic Sea. The primary objective of these measurements was to determine the geomagnetic field difference between the repeat stations and a designated reference geomagnetic observatory. Absolute geomagnetic observations were carried out at the RS sites during the morning
and evening hours (Csontos and Šugar 2024; Csontos 2023).

Given the considerable distance between the RS sites and the nearest observatories, additional efforts were required to improve the reliability of the collected data. To reduce the influence of external geomagnetic disturbances, a local magnetometer was installed on-site near the RS locations. This setup enabled the determination of field differences during quiet night-time periods, when external influences were minimal.

The employed instrument, a DIDD (Delta Inclination Delta Declination) system, provided the technological capability for on-site calibration. The instrument's performance, data processing methodology, and accuracy achieved during the campaign have been detailed in several previous publications Csontos et al. 2012; Kovács et al. 2012; Šugar et al. 2015). In addition, other studies investigated the geophysical manifestation of the so-called sea-side effect and discussed its relation to the induced current systems (Csontos 2013; Vujić and Brkić 2016).

The main conclusion of these investigations was the confirmed presence of an anomalous induced current system near the coastal region. However, the approaches adopted across these studies varied: while some focused on the appearance of geomagnetic gradient instabilities, others concentrated on the determination of the induced geomagnetic field. To achieve a more comprehensive understanding of the geomagnetic signature of the induction effect, a comparative analysis of both approaches is warranted.

For the purpose of the present study, data from two repeat stations (considered as anomalous points) and two geomagnetic observatories (serving as reference points) were analysed. The locations of these sites are shown in Figure 4.



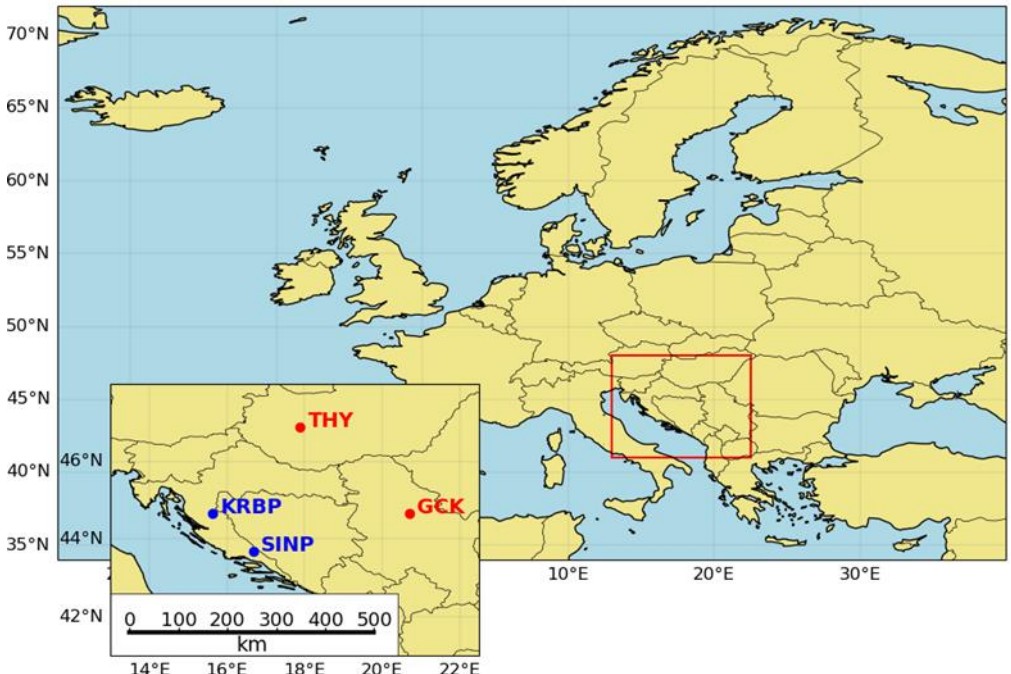

**Figure 4.: Geographic location of the repeat stations—Sinjsko Polje ([SINP] 21 km from the shoreline) and Krbavsko Polje**
**([KRBP] 53 km inland)—along with the reference geomagnetic observatories: Tihany ([IAGA code: THY]) and Grocka ([IAGA code: GCK]), indicated with red markers. The presence of the sea-side effect was confirmed at the repeat stations.**

### 3.1 Determination of the Parkinson vectors

The detection and separation of anomalous induced magnetic fields is based on the transfer function method, which
describes the linear relationship between the inducing normal geomagnetic field (denoted as $\mathbf{B_n}$ assumed to be homogeneous) and the induced anomalous magnetic field ($\mathbf{B_{ia}}$ considered inhomogeneous). By applying several simplifying assumptions (not detailed here) that are supported by empirical data sets, a simplified transfer function can be formulated [16, 2, 17, 7]:

$$Z'_{ia} = AX' + BY' + \varepsilon, \tag{19}$$

where $Z'_{ia}$ is the difference between the complex Fourier-transformed values of vertical components calculated as the
difference between the anomalous (costal) point and the reference observatory, A and B are complex coefficients of the simplified transfer function, and $\varepsilon$ is the residual of $Z'_{ia}$ that is not correlated to X' and Y' (variation of horizontal components on the normal point), i.e. instrumental noise and the neglected correlation with $Z_n$.(variation of vertical component on the normal point).





The solution of the Equation (19) is valid in the frequency domain. The generalized vector [A(ω), B(ω)] is known as the
induction vector (so called Parkinson arrow). The real part of the vector represents the in-phase response to the variations of
the horizontal field which can be define with amplitude and azimuth given by Equations (20) and (21), respectively:

$$\text{Amp}_r = \sqrt{|\Re(A)|^2 + |\Re(B)|^2},$$

(20)

$$\text{Az}_r = \arctan\left(\frac{\Re(A)}{\Re(B)}\right),$$

(21)

.

In an analogous manner, the quadrature (out-of-phase) response is obtained from the imaginary part of the vector [A(ω),
B(ω)]. The direction of the quadrature vector is theoretically expected to be opposite to that of the in-phase vector, i.e.,
pointing towards the source of the induced magnetic field (Banks 1973; Viljanen et al. 1995; Vujić and Brkić 2016).

It is well known that the intensity of the induction effect increases during geomagnetically disturbed periods (e.g., magnetic
storms). However, the strength of the anomalous magnetic field also depends on a variety of additional factors, including the
salinity, temperature, and depth of the sea. These physical parameters influence the electrical conductivity contrast between
the sea and the mainland and contribute to the generation of induced currents via motional induction and/or variations in the
geomagnetic field.

Importantly, sea-side effects have also been observed under geomagnetically quiet conditions. In such cases, the method of
"plane wave event" identification has been introduced as a selection criterion for reliable analysis (Viljanen et al. 1995;
Vujić and Brkić 2016). The criteria include:

• a strong linear correlation between the horizontal components recorded at the reference and anomalous stations, and

• a moderate relative difference (less than ~25%) between the horizontal variations at the anomalous site and the reference
site.

After identifying periods that satisfy these conditions, the selected geomagnetic time series can be subjected to robust cross-
spectral Fast Fourier Transform (FFT) analysis. This process yields the transfer function components A(ω) and B(ω) by
solving Equation (19).

**3.2 Results of classical induction arrow determination**

In the cited work of Vujić and Brkić 2016, two relevant approaches are presented for evaluating the results of classical
induction arrow analysis. The first is a tabular representation (see Table 1), where the directions and corresponding azimuths
of the in-phase responses are provided. The results indicate that the in-phase components of the induction vectors were
clearly and consistently determined for both reference observatories (GCK and THY).

**Table 1.** Results of the transfer functions analysis for KRBP and SINP, using observatories GCK and THY as a reference
point. All entries are written as mean±standard deviation. Amp and Az are the lengths and azimuths of induction arrows,
indices (r) apply to the real arrows. The values refer to the average and the standard deviation of data obtained from periods



of (10.7 min, 12.8 min, 18.3 min, 25.6 min, 32.0 min and 64.0 min) for KRBP, and of (12.8 min, 18.3 min, 21.3 min, 42.7 min and 64.0 min) for SINP. (After Vujić and Brkić 2016)

| | $Amp_r$ | $Az_r/°$ | $Amp_r$ | $Az_r/°$ |
|---|---|---|---|---|
| **Repeat station** | GCK | | THY | |
| KRBP | 0.30 ±0.07 | 266 ±17 | 0.35 ±0.07 | 259 ±10 |
| SINP | 0.37 ±0.09 | 233 ±8 | 0.34 ±0.14 | 234 ±8 |

In addition to the tabular data, the induction arrows were also graphically represented for each identified coastal effect period (see Figures 5 and 6). In these figures, the real components (in-phase responses) are depicted in black, while the imaginary components (quadrature responses) are shown in red.

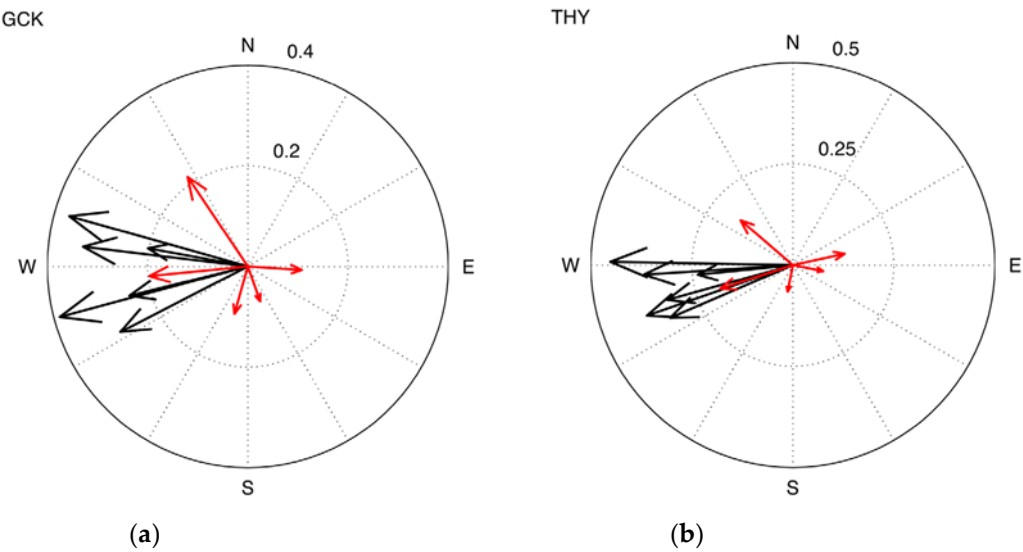

(a)    (b)

**Figure 5.: Real (black) and imaginary (red) induction arrows for KRBP, with GCK (a) and THY (b) as reference points. These arrows have been obtained for periods of 10.7 min, 12.8 min, 18.3 min, 25.6 min, 32.0 min and 64.0 min (after Vujić and Brkić 2016).**




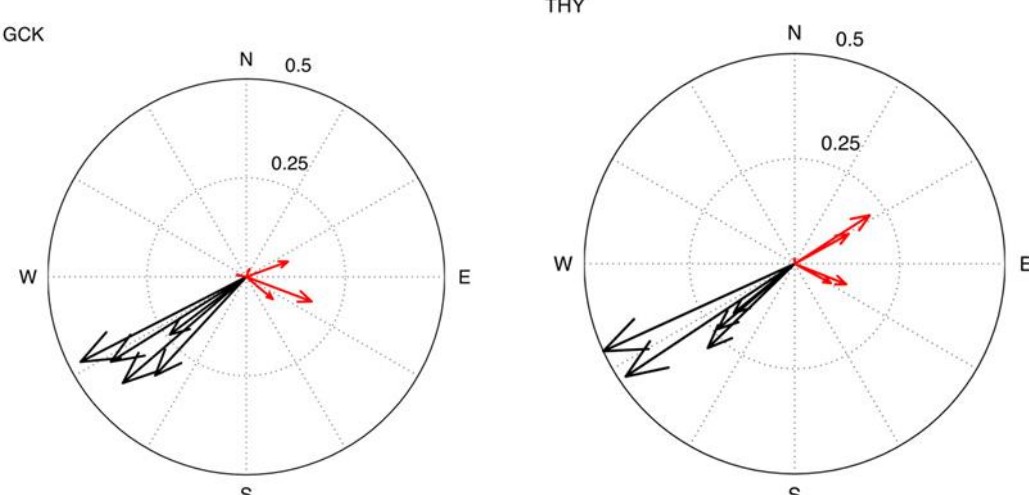


**Figure 6.: Real (black) and imaginary (red) induction arrows for SINP, with GCK (a) and THY (b) as reference points. These arrows have been obtained for periods of 12.8 min, 18.3 min, 21.3 min, 42.7 min and 64.0 min (after Vujić and Brkić 2016).**

The relatively small standard deviations observed for the amplitudes of the real induction arrows can be interpreted as

indicative of a stable sea-side induction effect. These previously published results are adopted as reference benchmarks for comparison with the findings presented in the subsequent sections of this study.

**4 Determination of induction vector by applying measured geomagnetic gradient**

This section presents the practical application of Equations (15), (16), (17), and (18) for the determination of the horizontal geomagnetic gradient and its directional characteristics. To carry out these calculations, the input parameters must be clearly defined. The magnitude of $W_H$ depends on the scalar value of the real offset ($S_0$) which is assumed to be relatively stable over the few-day timespan of the repeat station reoccupation. In contrast, the measured offsets $S_I$ and $S_D$, influenced by the activity of anomalous current systems, exhibit independent fluctuations across a wide range. By definition, the value of $W_H$ is

always positive. To estimate a suitable value for $S_0$, we used the criterion that the average of all $W_H$ values measured at the KRBP and SINP stations should be minimized. This condition was satisfied when $S_0$=2.75 nT.

The parameter $p$, which accounts for the ratio of sensor distances in declination and inclination measurements, is derived from the geometrical properties of the DIM device. Although direct measurement of this parameter is prone to uncertainty, the application of $p$=2.3 yielded consistent azimuthal values $Az_W$ during intervals characterized by elevated $W_H$ amplitudes.

This value is deemed reasonable based on the geometric interpretation shown in Fig. 2.

The change of the geometrical structure of the anomalous geomagnetic field maybe not linear in the in spatial volume. In this case p is a technical parameter.



The intervals identified as "plane wave events" are taken from the earlier study (Vujić and Brkić 2016) and are summarized in Table 2 along with the corresponding DIM measurement periods. When simultaneous DIM data were unavailable, DIM records from the same day—prioritizing those with high $W_H$ amplitudes—were used.

Table 2. Selected time intervals of plane wave events on July 21 and 22, 2010, for KRBP, and July 26, 2010, for SINP and corresponding DIM gradient measurement windows.

| | Transfer function | | Gradient | |
|---|---|---|---|---|
| *Date* | *Time interval (UTC)* | | *Time interval (UTC)* | |
| 2010-07-21 | 15:31:00 | 17:39:00 | 08:28:00 | 08:36:00 |
| 2010-07-21 | 15:39:00 | 17:47:00 | 17:08:00 | 17:15:00 |
| 2010-07-21 | 15:55:00 | 18:03:00 | 17:21:00 | 17:28:00 |
| 2010-07-22 | 16:10:00 | 18:18:00 | 17:04:00 | 17:14:00 |
| 2010-07-22 | 16:31:00 | 18:39:00 | 16:37:00 | 16:45:00 |
| 2010-07-22 | 16:51:00 | 18:59:00 | 16:48:00 | 16:57:00 |
| 2010-07-22 | 14:07:00 | 16:15:00 | 14:42:00 | 14:48:00 |
| 2010-07-26 | 08:09:00 | 10:17:00 | 06:10:00 | 06:18:00 |
| 2010-07-26 | 08:34:00 | 10:42:00 | 06:24:00 | 06:32:00 |
| 2010-07-26 | 08:37:00 | 10:45:00 | 06:36:00 | 06:46:00 |
| 2010-07-26 | 08:39:00 | 10:47:00 | 17:03:00 | 17:12:00 |
| 2010-07-26 | 17:27:00 | 19:35:00 | 17:15:00 | 17:24:00 |
| 2010-07-26 | 17:42:00 | 19:50:00 | 17:27:00 | 17:35:00 |
| 2010-07-26 | 17:46:00 | 19:54:00 | 17:40:00 | 17:51:00 |

To visualize the results obtained via the gradient-based method, the amplitude and strike direction of the horizontal magnetic gradient were determined and are presented as vectors in Figure 7.

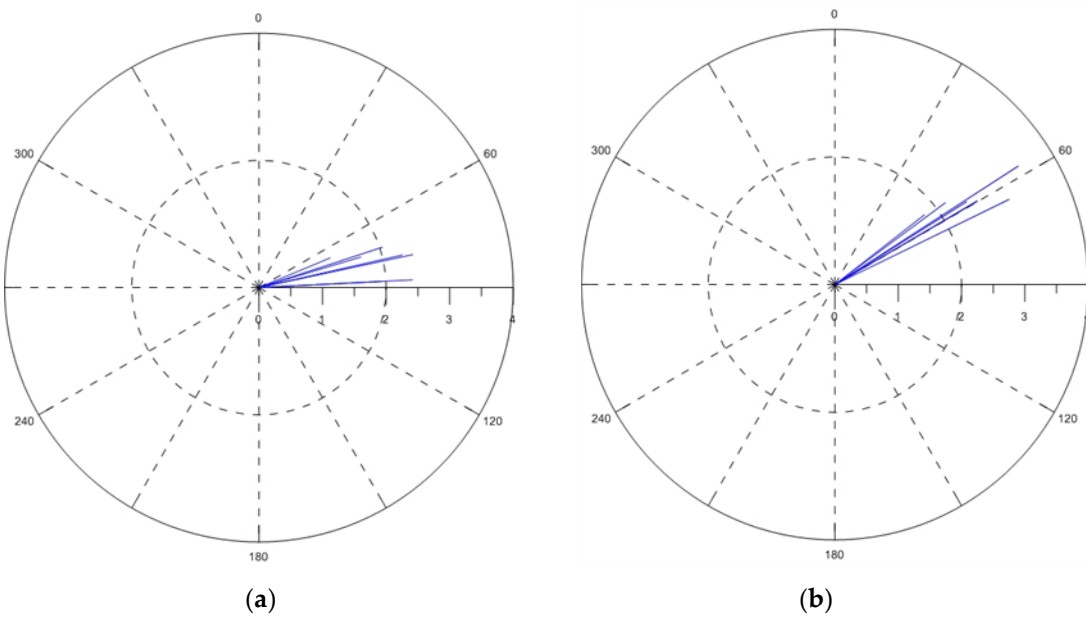

(a)                                    (b)



**Figure 7.: Gradient vectors derived from DIM measurements at KRBP RS (a) and SINP RS (b) using the time intervals listed in Table 2.**

In order to create more systematic comparison of the methods the mean values of the amplitudes and the azimuths was calculated too.

The ratio of the amplitudes presents that the observed horizontal gradient of geomagnetic field was higher with 30% on SINP RS which is located closer to the sea. This result is between the previously determined ratios (see Table 1. or Fig. 5. and Fig. 6. for the details). The explanation of the difference can be that the gradient observation measures the superposition

of anomalous currents of the area. Moreover the difference between the presented values of the different methods is not significant. The relatively high scatter of the amplitude is a consequence of gradient instability.

Despite observable scatter in amplitude values—attributable to gradient instability—the directional consistency remains notable. When accounting for a 180° ambiguity in azimuth interpretation, the derived Parkinson vector directions are 257° at KRBP RS and 238° at SINP RS. These differ by only 2° and 4°, respectively, from those obtained via the transfer function

method (Table 1). Furthermore, the standard deviations of the azimuth values derived from the gradient method are approximately half of those observed in the classical approach, indicating improved directional stability.

**5 Discussion**

The results demonstrate that the geomagnetic gradient is a suitable and sensitive parameter for detecting the presence of

anomalous, localized current systems. In particular, the Krbavsko Polje repeat station (RS), located more than 50 km inland from the Adriatic Sea, still showed measurable effects attributed to the coastal induction phenomenon. Notably, previous studies have reported that the sea-side effect can be detected at distances exceeding 150 km inland during geomagnetic storms (Viljanen et al. 1995; Arora et al. 1999; Srivastava et al. 2001). This finding highlights the potential of using geomagnetic gradient measurements not only for coastal effects but also for detecting the influence of subsurface anomalous

currents.

In cases involving underground current systems, it is expected that the vertical component of the geomagnetic gradient—defined in Equation (6)—would be predominantly affected (Csontos 2019). Furthermore, there is no a priori requirement for the measured geomagnetic gradient variations to be solely driven by external field changes. For example, transient increases in the electrical conductivity of the Earth's crust—often observed prior to seismic events—have been widely studied as

potential precursors to earthquakes (Han et al. 2021).

Interestingly, the observed geomagnetic gradients associated with anomalous currents challenge the conventional assumption that magnetic anomalies decay rapidly with distance. Earlier studies (Pedersen and Rasmussen 1990) focused on the spatial geometry of magnetic anomalies produced by localized dipole sources. In contrast, anomalies induced by large-scale current systems often exhibit a solenoidal structure, leading to more spatially extended magnetic gradients. Although the strongest

gradients typically occur within small spatial volume, local fluctuations in the differences between the corresponding components of the geomagnetic field have been observed across all recorded data.

Moreover, the potential for detecting secondary electromagnetic (EM) waves generated by anomalous current systems is also suggested by these results. Until now, the application of magnetic gradiometers in EM induction studies has primarily been limited to resource exploration involving artificially generated primary EM fields (Veryaskin 2018). These new observations

highlight the effectiveness of using gradient-based magnetic measurements in detecting naturally occurring geomagnetic induction phenomena.

Altogether, these findings underscore the need for continued investigation to further characterize and understand the spatial and temporal dynamics of geomagnetic gradients associated with both natural and anomalous sources.

## 6 Conclusions

This study presented a comparative analysis of two distinct methods used to quantify the sea-side effect observed during the reoccupation of two geomagnetic repeat stations located near the Adriatic Sea. A novel approach for measuring the horizontal geomagnetic gradient was introduced, utilizing a DIM (Declination-Inclination Magnetometer) instrument.

The results obtained from the gradient-based method—including both the intensity and the strike direction of the horizontal magnetic gradient—were in strong agreement with those derived from the established transfer function approach,

specifically the determination of the Parkinson vectors. This concordance confirms the validity and reliability of the proposed gradient-based method.

The geomagnetic gradient method offers a transparent, efficient, and practical framework for quantifying induction phenomena, particularly in situ and in near real-time. In addition to capturing the influence of coastal induction effects, this method also demonstrates the potential to detect the presence and activity of subsurface (beneath) current systems from

considerable distances.

These findings underscore the broader applicability of the geomagnetic gradient technique, not only as a complement to traditional transfer function analysis but also as a stand-alone diagnostic tool for monitoring spatially extended induction effects in both marine and continental environments.

**Acknowledgement**

For the illustrative figures (1 and 4) András Csontos junior is acknowledged. ChatGPT was used for correcting grammar in the text.

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
