# Peer review of "Characterizing anomalous geomagnetic induction from coastal effects with transfer functions and gradient measurements"

_EGUsphere, 2025_

## Referee Comment (RC1)

Review on the manuscript

**_Characterizing anomalous geomagnetic induction from coastal effects with transfer functions and gradient measurements_**
_by András Attila Csontos_

The author of the manuscript presents an idea of evaluating the anomalous electromagnetic induction, which manifests itself in the measured elements of the geomagnetic field, through the gradient of the geomagnetic field obtained in a location with extreme geomagnetic disturbance. In order to give this idea scientific depth, first in Section 1/Introduction he proposes a mechanism for the formation of disturbances at the interface of two conductivity-contrasting environments. In Section 2 he presents his approach to performing geomagnetic measurements using a Declination-Inclination magnetometer (DIM), which are intended to serve to evaluate the gradient of the geomagnetic field. In Section 3 he presents the circumstances of the measurements at 2 repeat stations near the Adriatic Sea and mentions the analysis of the geomagnetic coastal effect using the transfer functions of the electromagnetic field by the authors (Vujić and Brkić, 2016). Section 4 is devoted to his own contribution, which is discussed in Section 5 and concluded in Section 6.

In summary: Aside from the technical side of the matter, this manuscript does not represent a coherent, self-supporting text and the understanding of the author's idea depends too much on other literature. In this regard, I would like to note that the scope of the material taken from the work (Vujić and Brkić, 2016) is disproportionately large and gives the impression that it is part of the presented concept. Thus, when attempting to publish this manuscript, it encounters an ethical issue.

As for the scientific substance of the manuscript, the very idea of performing absolute measurements of the geomagnetic field in a location where it is extremely disturbed does not make sense. Moreover, this idea could not be fulfilled in principle by the presented approach, since (1) the angular measurements using DIM (declination and inclination measurements) were not combined with the total field measurements using a scalar magnetometer and (2) the angular measurements using DIM could not be realized by the procedure described in the relevant section.

The author does not state the results of the alleged measurements.

The origin of the main outcome in the form of Figure 7 is unknown and therefore it is not possible to comment on it.

Conclusion: This manuscript does not represent a scientific study and no part of its text can be part of a future scientific publication.

I explain my reasons in more detail below.

- One of the important assumptions of the study (Vujić and Brkić, 2016), on which the presented manuscript is based, is the fact that in mid-latitudes the excitation field can be considered as a plane wave propagating vertically downwards. This is an important assumption that has a physical justification and which allowed the analysis to be carried out using transfer functions in a significantly simplified form. In such a wave, the electric and magnetic field vectors oscillate in the horizontal plane. This means that the magnetic flux through the sea surface is zero and therefore the configuration of fields $\mathbf{B}$ and $\mathbf{j}$, as

drawn in Figure 1 presented in Introduction, cannot occur.

Even if we assume another external source of excitation field, which is capable of inducing currents similar to those depicted in the figure and which superimpose into a line current at the interface of the environments, as depicted, then it should be noted that the magnitude of the magnetic field of such a current decreases with the square of the distance (Biot-Savart-Laplace law). I recommend the author to make estimates of the relevant quantities based on the equations and to assess the magnitude of the resulting effect at distances of measurement sites ($O(10^4)$ m) from the sea.

This idea is therefore unable to physically explain the anomalous variation of the $Z$-component of the geomagnetic field.

- I would expect that in Section 2, dedicated to the actual measurement using DIM, the author would briefly state its principle. Not only does he not state it, but he does not even follow it, which raises serious doubts about his declared professional competence.

  First, the declination must be measured, not only for the sake of the value sought, but also to determine the meridional plane in which the inclination is subsequently measured. However, without giving reasons and further details, the author begins by determining the inclination (and other quantities whose purpose is unclear). Without further explanation and future use, he states the relationship for the variation of the magnetic field in the vertical direction, into which the value of the magnitude of the total field enters, which, however, was not the subject of the measurements. The culmination of this theoretical part is the determination of the DIM offset, i.e. the systematic error of the instrument, which slightly modifies the measurement results. The use of this offset and misalignment errors in the horizontal and vertical planes for determining the sought-after characteristics (geomagnetic field gradient) and the formal correctness of the formulas for them will not be commented on for obvious reasons.

  No values of measured declination and inclination are given.

- The entire Section 3 is an excerpt from the work (Vujić and Brkić, 2016) without any own contribution.

- There are no facts provided on the basis of which it would be possible to comment on the result in the form of Figure 7 presented in Section 4. For the same reason, I will not comment on Sections 5 and 6.

| | |
|---|---|
| Scientific significance: Does the manuscript represent a substantial contribution to scientific progress within the scope of Geoscientific Instrumentation, Methods and Data Systems (substantial new concepts, ideas, methods, or data)? | No (4) |
| Scientific quality: Are the scientific approach and applied methods valid? Are the results discussed in an appropriate and balanced way (consideration of related work, including appropriate references)? | No (4) |
| Presentation quality: Are the scientific results and conclusions presented in a clear, concise, and well-structured way (number and quality of figures/tables, appropriate use of English language)? | Poor (4) |

---

## Author Comment (AC1)

**The author's responses to the anonymous reviewer's comments.**

For better clarity, I have divided my responses into two groups. In the first group, I respond to the reviewer's suggestions that are suitable for scientific discourse. In the second group, I also added my comments on the untrue statements, professional absurdities, and unidentified text sections in the review.

The sentences in the review are in italics without any changes. The answers are in normal font.

**General respectable remarks:**

*Aside from the technical side of the matter, this manuscript does not represent a coherent, self-supporting text and the understanding of the author's idea depends too much on other literature.*

The text indeed assumes prior knowledge and expects the reader to thoroughly review the cited articles. Perhaps I was overly ambitious when I sought to address two methodologically different approaches, along with their geophysical mechanism and measurement innovation, in a single study, which would allow the conclusions to emerge.

*In this regard, I would like to note that the scope of the material taken from the work (Vujić and Brkić, 2016) is disproportionately large and gives the impression that it is part of the presented concept. Thus, when attempting to publish this manuscript, it encounters an ethical issue.*

First, I must note that I intended to prepare a comparative study, and I clearly stated this intention in the abstract. In such a case, results previously published in a scientific article can strengthen the feasibility of the goal, as they provide a verified reference for checking the newly presented method. If I shorten this section, the method for determining the Parkinson vector will no longer be comprehensible.

Second, I am puzzled by the ethical concern, because, just as in this study, the earlier publications (*Vujić and Brkić, 2016 also*) mostly analyzes my own measurements. This fact is also clearly evident from the cited studies.

*The author does not state the results of the alleged measurements.*

It seemed unnecessary to repeat these results in the manuscript, as all of them have already been published (Csontos and Šugar 2024).

*The origin of the main outcome in the form of Figure 7 is unknown and therefore it is not possible to comment on it.*

The numerical results can be obtained from the previously published data using the formulas discussed in detail. If this is indeed considered necessary, I can provide them in the form of two simple tables (see below).

| Time (2010) | | | $D_{abs}$ | | $I_{abs}$ | | Offsets | | $W_H$ | |
|---|---|---|---|---|---|---|---|---|---|---|
| DoY | hh | mm | (°) | (') | (°) | (') | $S_{0D}$ (nT) | $S_0$ (nT) | (rel. scale) | (°) |
| 201 | 5 | 14 | 2 | 53,62 | 61 | 13,75 | 3,966 | 4,81 | 1,59 | -86,24 |
| 201 | 5 | 32 | 2 | 53,67 | 61 | 13,67 | 4,958 | 3,78 | 2,27 | -89,99 |
| 201 | 5 | 42 | 2 | 53,72 | 61 | 13,85 | 3,966 | 3,44 | 1,26 | -90,67 |
| 201 | 6 | 2 | 2 | 54,17 | 61 | 14,15 | 4,626 | 4,81 | 2,14 | -86,83 |
| 201 | 6 | 20 | 2 | 54,12 | 61 | 14,45 | 3,304 | 4,12 | 0,88 | -87,96 |
| 201 | 6 | 38 | 2 | 54,00 | 61 | 14,83 | 4,459 | 4,47 | 1,91 | -87,71 |
| 201 | 16 | 20 | 2 | 49,57 | 61 | 14,60 | 1,321 | 3,43 | 1,47 | -89,17 |
| 201 | 16 | 39 | 2 | 49,72 | 61 | 14,50 | 2,973 | 2,06 | 0,41 | 84,54 |
| 201 | 17 | 0 | 2 | 49,40 | 61 | 14,43 | 0,165 | 3,09 | 2,59 | -90,26 |
| 201 | 17 | 13 | 2 | 49,77 | 61 | 14,58 | 1,982 | 3,09 | 0,79 | -91,26 |
| 201 | 17 | 30 | 2 | 49,82 | 61 | 14,43 | 2,313 | 2,40 | 0,47 | 85,66 |
| 201 | 17 | 45 | 2 | 49,92 | 61 | 14,40 | 2,973 | 2,75 | 0,22 | 87,16 |
| 202 | 8 | 2 | 2 | 54,70 | 61 | 14,63 | 2,807 | 3,09 | 0,18 | -91,58 |
| 202 | 8 | 15 | 2 | 54,40 | 61 | 14,33 | 1,817 | 1,72 | 1,06 | 82,16 |
| 202 | 8 | 33 | 2 | 53,85 | 61 | 14,17 | 0,496 | 1,72 | 2,31 | 80,18 |
| 202 | 8 | 48 | 2 | 53,45 | 61 | 14,05 | 1,156 | 0,69 | 1,89 | 75,72 |
| 202 | 17 | 13 | 2 | 50,35 | 61 | 14,60 | 0,826 | 1,37 | 2,04 | 78,81 |
| 202 | 17 | 26 | 2 | 50,72 | 61 | 14,53 | 0,33 | 1,72 | 2,47 | 79,86 |
| 202 | 17 | 48 | 2 | 51,00 | 61 | 14,43 | 1,156 | 1,03 | 1,81 | 77,62 |
| 202 | 18 | 0 | 2 | 50,87 | 61 | 14,43 | 1,321 | 0,34 | 1,86 | 74,45 |
| 203 | 7 | 15 | 2 | 53,85 | 61 | 16,70 | 3,134 | 2,75 | 0,38 | 87,10 |
| 203 | 7 | 27 | 2 | 53,92 | 61 | 15,78 | 2,641 | 3,09 | 0,20 | -91,51 |
| 203 | 7 | 45 | 2 | 53,72 | 61 | 15,95 | 2,64 | 2,75 | 0,11 | 87,10 |
| 203 | 7 | 58 | 2 | 54,52 | 61 | 15,97 | 2,97 | 1,03 | 0,88 | 80,55 |
| 203 | 8 | 21 | 2 | 52,45 | 61 | 16,00 | 2,145 | 0,69 | 1,19 | 78,00 |
| 203 | 14 | 35 | 2 | 49,12 | 61 | 14,55 | 2,312 | 2,06 | 0,56 | 84,21 |
| 203 | 14 | 47 | 2 | 49,25 | 61 | 14,63 | 1,156 | 1,72 | 1,67 | 81,42 |
| 203 | 16 | 42 | 2 | 51,70 | 61 | 14,95 | 0,826 | 2,75 | 1,92 | 87,13 |
| 203 | 16 | 55 | 2 | 51,52 | 61 | 15,25 | 0,33 | 2,75 | 2,42 | 87,12 |
| 203 | 17 | 10 | 2 | 51,62 | 61 | 15,17 | 1,651 | 1,72 | 1,21 | 82,02 |
| 203 | 17 | 25 | 2 | 51,67 | 61 | 15,00 | 2,312 | 2,06 | 0,56 | 84,16 |
| 203 | 17 | 38 | 2 | 51,47 | 61 | 14,98 | 1,982 | 2,40 | 0,79 | 85,55 |
| 203 | 18 | 10 | 2 | 50,85 | 61 | 14,83 | 2,807 | 2,40 | 0,18 | 85,79 |
| 204 | 5 | 41 | 2 | 54,95 | 61 | 14,15 | 2,479 | 2,75 | 0,27 | 87,06 |
| 204 | 5 | 52 | 2 | 54,82 | 61 | 14,18 | 2,313 | 3,09 | 0,47 | -91,44 |
| 204 | 6 | 3 | 2 | 54,85 | 61 | 14,28 | 2,478 | 3,78 | 0,58 | -88,63 |
| 204 | 6 | 15 | 2 | 55,00 | 61 | 13,98 | 2,479 | 3,78 | 0,58 | -88,63 |
| 204 | 6 | 31 | 2 | 55,32 | 61 | 14,18 | 1,983 | 3,78 | 0,92 | -88,18 |

*Results of DIM measurements at KRBP repeat station and the calculated horizontal gradient ($W_H$) amplitudes and their strike directions. Values observed during the time periods selected for spectral analysis are highlighted in red.*

| Time (2010) | | | $D_{abs}$ | | $I_{abs}$ | | Offsets | | $W_H$ | |
|---|---|---|---|---|---|---|---|---|---|---|
| DoY | hh | mm | (°) | (') | (°) | (') | $S_{0D}$ (nT) | $S_0$ (nT) | (rel. scale) | (°) |
| 204 | 18 | 7 | 2 | 51,62 | 60 | 28,62 | 1,18 | 1,71 | 1,65 | 68,85 |
| 204 | 18 | 18 | 2 | 51,15 | 60 | 28,62 | 2,03 | 1,71 | 0,89 | 51,57 |
| 204 | 18 | 29 | 2 | 51,32 | 60 | 28,58 | 0,84 | 1,03 | 2,09 | 62,84 |
| 205 | 6 | 1 | 2 | 56,12 | 60 | 29,08 | 2,53 | 3,77 | 0,55 | -26,21 |
| 205 | 6 | 11 | 2 | 56,32 | 60 | 29,08 | 1,52 | 3,08 | 1,24 | -85,25 |
| 205 | 6 | 22 | 2 | 56,35 | 60 | 29,13 | 1,35 | 3,08 | 1,41 | -86,17 |
| 205 | 6 | 35 | 2 | 56,37 | 60 | 29,23 | 2,19 | 1,71 | 0,76 | 44,09 |
| 205 | 6 | 48 | 2 | 56,32 | 60 | 29,28 | 1,52 | 1,71 | 1,34 | 64,24 |
| 205 | 7 | 3 | 2 | 56,12 | 60 | 29,20 | 2,19 | 2,06 | 0,66 | 55,10 |
| 205 | 17 | 17 | 2 | 53,22 | 60 | 28,88 | 1,52 | 3,08 | 1,24 | -85,20 |
| 205 | 17 | 31 | 2 | 52,80 | 60 | 28,98 | 3,71 | 2,40 | 0,98 | -82,51 |
| 205 | 17 | 49 | 2 | 52,47 | 60 | 29,03 | 3,54 | 3,08 | 0,81 | 75,30 |
| 205 | 18 | 4 | 2 | 52,70 | 60 | 29,25 | 4,05 | 3,43 | 1,34 | 72,57 |
| 205 | 18 | 23 | 2 | 52,30 | 60 | 29,13 | 3,37 | 3,08 | 0,65 | 72,23 |
| 206 | 17 | 27 | 2 | 53,25 | 60 | 29,68 | 5,40 | 5,82 | 3,06 | 57,01 |
| 206 | 17 | 47 | 2 | 53,25 | 60 | 29,40 | 2,36 | 4,80 | 1,09 | -23,66 |
| 206 | 18 | 2 | 2 | 52,85 | 60 | 29,35 | 2,70 | 5,48 | 1,37 | -4,99 |
| 206 | 18 | 14 | 2 | 53,15 | 60 | 29,45 | 3,04 | 5,48 | 1,39 | 8,99 |
| 207 | 6 | 16 | 2 | 56,05 | 60 | 29,98 | 5,06 | 5,14 | 2,60 | 59,76 |
| 207 | 6 | 29 | 2 | 56,22 | 60 | 30,02 | 5,57 | 5,14 | 3,06 | 64,11 |
| 207 | 6 | 44 | 2 | 55,45 | 60 | 30,05 | 5,73 | 6,17 | 3,44 | 57,32 |
| 207 | 6 | 59 | 2 | 55,65 | 60 | 30,20 | 4,72 | 4,11 | 2,09 | 68,06 |
| 207 | 7 | 18 | 2 | 56,25 | 60 | 30,30 | 3,03 | 4,11 | 0,74 | 19,83 |
| 207 | 17 | 9 | 2 | 52,57 | 60 | 29,58 | 4,89 | 5,14 | 2,45 | 58,01 |
| 207 | 17 | 21 | 2 | 52,70 | 60 | 29,58 | 5,06 | 5,14 | 2,60 | 59,82 |
| 207 | 17 | 32 | 2 | 52,57 | 60 | 29,50 | 4,22 | 4,80 | 1,79 | 52,28 |
| 207 | 17 | 48 | 2 | 52,22 | 60 | 29,53 | 4,55 | 5,14 | 2,16 | 53,67 |
| 207 | 18 | 7 | 2 | 52,05 | 60 | 29,50 | 3,37 | 4,80 | 1,20 | 28,55 |

*Results of DIM measurements at SINP repeat station and the calculated horizontal gradient ($W_H$) amplitudes and their strike directions. Values observed during the time periods selected for spectral analysis are highlighted in red.*

*Even if we assume another external source of excitation field, which is capable of inducing currents similar to those depicted in the figure and which superimpose into a line current at the interface of the environments, as depicted, then it should be noted that the magnitude of the magnetic field of such a current decreases with the square of the distance (Biot-Savart-Laplace law). I recommend the author to make estimates of the relevant quantities based on the equations and to assess the magnitude of the resulting effect at distances of measurement sites ($O(10^4)$ m) from the sea. This idea is therefore unable to physically explain the anomalous variation of the Z-component of the geomagnetic field.*

I acknowledge that figure is indeed lacking in that it is unable to reflect the induction effects appearing at different frequency ranges and their spatial propagation. In the revised manuscript, I will clarify its conceptual purpose and, where feasible, provide an additional figure illustrating these effects.

*I would expect that in Section 2, dedicated to the actual measurement using DIM, the author would briefly state its principle.*

I have provided the necessary references here, including an OA article *(Csontos and Šugar 2024)* among them. If I were to go into deeper detail, I would have to engage in discussions deviating from the manuscript's purpose.

*The entire Section 3 is an excerpt from the work (Vujić and Brkić, 2016) without any own contribution.*

True, but I have already addressed this earlier, since it is a comparative study.

**Professionally unfounded statements:**
*As for the scientific substance of the manuscript, the very idea of performing absolute measurements of the geomagnetic field in a location where it is extremely disturbed does not make sense.*

As long as artificial disturbances do not affect the data, the installation of repeat station points in anomalous regions is entirely acceptable. In the present case, as confirmed by the calculations, we recorded natural geomagnetic variation. Moreover, geomagnetic observatories also operate on volcanic islands (e.g., Tristan da Cunha [TDC]).

*Moreover, this idea could not be fulfilled in principle by the presented approach, since*

*(1) the angular measurements using DIM (declination and inclination measurements) were not combined with the total field measurements using a scalar magnetometer*

The truth is quite the opposite, as is clearly evident from the cited data publication (Csontos, A. A.: "DIM_sets", *Mendeley Data*, V2, doi: 10.17632/hzphnd3p42.2 2023).

*(2) the angular measurements using DIM could not be realized by the procedure described in the relevant section.*

The statement is ambiguous, and it is unclear what it refers to, but it seems to suggest some sort of failure. In reality, all studied DIM measurements have already been published *(Csontos and Šugar 2024)*, and therefore were successful.

*One of the important assumptions of the study (Vujić and Brkić, 2016), on which the presented manuscript is based, is the fact that in mid-latitudes the excitation field can be considered as a plane wave propagating vertically downwards. This is an important assumption that has a physical justification and which allowed the analysis to be carried out using transfer functions in a significantly simplified form. In such a wave, the electric and magnetic field vectors oscillate in the horizontal plane. This means that the magnetic flux through the sea surface is zero and therefore the configuration of fields B and j, as drawn in Figure 1 presented in Introduction, cannot occur.*

The figure represents the total inducing field $B_0$, which is the sum of the Earth's main field and the EM waves arising in the external regions. Furthermore, the skin depth is not zero even for seawater, as it is not a perfect conductor. A current of 100,000 A parallel to the coastline is a realistic value and forms part of the anomalous current system. This, in turn, generates a solenoidal anomalous magnetic field.

*Not only does he not state it, but he does not even follow it, which raises serious doubts about his declared professional competence. First, the declination must be measured, not only for the sake of the value sought, but also to determine the meridional plane in which the inclination is subsequently measured.*

This statement is not correct. Moreover, the order proposed by the reviewer is also incorrect, since without determining the direction of the magnetic meridian, in the own reference frame of the theodolite, it is impossible to determine the declination. On the other hand, inclination can be measured without determining declination if the direction of the magnetic meridian has already been measured in the theodolite's own reference frame.

*However, without giving reasons and further details, the author begins by determining the inclination (and other quantities whose purpose is unclear). Without further explanation and future use, he states the relationship for the variation of the magnetic field in the vertical direction, into which the value of the magnitude of the total field enters, which, however, was not the subject of the measurements.*

Without a specific location reference, we can only guess that the reviewer is referring to a quantity already published earlier but mentioned here (since it falls within the manuscript's subject). If a more precise location reference were provided, I would see an opportunity for deeper discussion.

*The culmination of this theoretical part is the determination of the DIM offset, i.e. the systematic error of the instrument, which slightly modifies the measurement results.*

The sentence contains a fundamental error, because the offset (if it remains stable during the observation period) does not in any way modify the results of DIM measurements regarding to the geomagnetic components. However, the offset instability is due to the inhomogeneity of the Earth's magnetic field. I have also cited the previous study on this (*Gilbert and Rasson 1998*). However, if we were to agree with the reviewer, the DIM device would not be an absolute instrument. Fortunately, the reality is the exact opposite.

*The use of this offset and misalignment errors in the horizontal and vertical planes for determining the sought-after characteristics (geomagnetic field gradient) and the formal correctness of the formulas for them will not be commented on for obvious reasons.*

I am unable to add further comments to this remark by the reviewer, as it is a statement made without any reason or explanation.

*No values of measured declination and inclination are given.*

I agree with the reviewer that this is correct. This was not the topic of the manuscript. Incidentally, the full results of the DIM measurements have already been published in an OA article *(Csontos and Šugar 2024)*. This statement by the reviewer indicates a failure to review the fundamental references.

**Summary:**

I think that the discourse so far has largely not touched on the scientific theses originally set out in the manuscript. In this strange situation where I was dragged into by this criticism, I tried to do my best at every point where I could be constructive, although on several points my conscience and knowledge did not allow this. I think the assessment of the situation has to be left to the professional quality of the editors.